# Genome-wide genetic and epigenetic analyses of pancreatic acinar cell carcinomas reveal aberrations in genome stability

Cornelia Jäkel[1], Frank Bergmann[2], Reka Toth[1], Yassen Assenov[1], Daniel van der Duin[1], Oliver Strobel[3], Thomas Hank[3], Günter Klöppel[4], Craig Dorrell [5], Markus Grompe [5], Joshua Moss[6], Yuval Dor[6], Peter Schirmacher[2], Christoph Plass[1], Odilia Popanda[1] & Peter Schmezer[1]

Pancreatic acinar cell carcinoma (ACC) is an aggressive exocrine tumor with largely unknown biology. Here, to identify potential targets for personalized treatment, we perform integrative genome-wide and epigenome-wide analyses. The results show frequently aberrant DNA methylation, abundant chromosomal amplifications and deletions, and mutational signatures suggesting defective DNA repair. In contrast to pancreatic ductal adenocarcinoma, no recurrent point mutations are detected. The tumor suppressors *ID3*, *ARID1A*, *APC*, and *CDKN2A* are frequently impaired also on the protein level and thus potentially affect ACC tumorigenesis. Consequently, this work identifies promising therapeutic targets in ACC for drugs recently approved for precision cancer therapy.

[1] Division of Epigenomics and Cancer Risk Factors, German Cancer Research Center (DKFZ), Im Neuenheimer Feld 280, 69120 Heidelberg, Germany. [2] Institute of Pathology, University Hospital Heidelberg, Im Neuenheimer Feld 224, 69120 Heidelberg, Germany. [3] Department of General and Visceral Surgery, University Hospital Heidelberg, Im Neuenheimer Feld 110, 69120 Heidelberg, Germany. [4] Institute of Pathology, Technical University Munich, Trogerstr. 18, 81675 Munich, Germany. [5] Department of Pediatrics, Papé Family Pediatric Research Institute, Oregon Stem Cell Center, Oregon Health and Science University, Portland, OR 97239, USA. [6] Department of Developmental Biology and Cancer Research, Institute for Medical Research Israel-Canada, The Hebrew University-Hadassah Medical School, 9112102 Jerusalem, Israel. Correspondence and requests for materials should be addressed to P.S. (email: p.schmezer@dkfz.de)

Pancreatic acinar cell carcinomas (ACC) are aggressive cancers that most frequently affect patients in their sixth and seventh decades of life, but may occur at any age, including childhood. At the time of diagnosis, ~50% of ACC present with metastases[1–4]. Phenotypically, ACC resemble non-neoplastic acinar cells[1]. Rarely, ACC display a substantial neuroendocrine mixed differentiation, then being designated as mixed acinar-neuroendocrine carcinoma (MACNEC)[1]. Apart from tumor resection, which is the first line therapy in localized disease, therapeutic options in advanced disease are limited and have shown varying results[5]. New treatment strategies, including targeted drugs, are therefore needed. This is hampered by the fact that the molecular background of ACC is still only poorly understood, in part because it is a rare pancreatic neoplasm that accounts for <2% of all pancreatic cancers[4]. This is in contrast to (i) pancreatic ductal adenocarcinomas (PDAC) accounting for over 90% of pancreatic cancers, where molecular alterations, e.g., in KRAS, TP53, or CDKN2A are well known[6, 7] and (ii) pancreatic neuroendocrine tumors (PNET) that show frequent alterations, e.g., in MEN1, MUTYH, or CHEK2[8]. Several smaller studies revealed that these genes are only rarely affected in ACC[9–12]. Two recent genome-wide exome-sequencing studies demonstrated point mutations in ACC[13, 14], although no highly recurrent point mutations were detected, and the most common point mutation occurred in only 25% of the tumors in the SMAD4 gene[13, 14]. In a study using comparative genomic hybridization analysis, numerous chromosomal imbalances were detected[15]. Albeit, resolution of this array-based study is very limited thus making it difficult to pinpoint distinct important driver aberrations. Hence, no frequently recurrent point mutations or genetic driver events were found in ACC, suggesting other predominant mechanisms of tumor development.

Additional support for this assumption comes from findings that the Wnt pathway is affected in at least a subset of ACC. Mutations in APC have been reported in about 10–20% of cases[9, 11, 14] and activation of the CTNNB1 protein has been reported in about 12–15%[9, 16]. One study identified higher recurrence of aberrations by investigating APC on the level of mutations, deletions, and hypermethylation, and therefore detected changes in 7, 48, and 56% of cases, respectively[12]. However, data on the APC protein expression are still lacking.

In this study, we investigate two independent cohorts with a total number of 73 ACC and 34 normal pancreatic tissue samples representing one of the largest available tissue-based collectives world-wide. We aimed to examine the molecular aberrations in ACC in a comprehensive manner by investigating point mutations, DNA methylation, and copy number alterations employing whole-exome sequencing (WES) and Illumina's 450K array (450K). In summary, we show that ACC do not display recurrent point mutations, but exhibit distinct mutational signatures. They harbor a large number of gene deletions and differential methylation of promoter sites and genes. This may provide directions for targeted therapeutic interventions.

## Results

**Genetic profiles confirm no recurrent point mutations in ACC.** To investigate molecular alterations in ACC, whole exomes of ACC with available matching normal tissue were sequenced (22 ACC from cohort 1; see Supplementary Table 1). We obtained a median number of 137 point mutations per tumor with a range from 40 to 1023 mutations per tumor (Fig. 1a), similar to previous reports by Jiao et al.[14] and Furukawa et al.[13] (Fig. 1b, c). The high mutational load (>1000 mutations) of one tumor was explained by microsatellite instability of this tumor, previously reported in Bergmann et al.[15] (Supplementary Data

File 1). Genes mutated in 23% of carcinomas were COL12A1, FRY, FRYL, and PLB1 followed by CACNA1A, CCDC57, COL23A1, MKL2, RAP1GAP, SMAD4, and SRCAP occurring in 18% of carcinomas (Fig. 1b, for a detailed list of point mutations refer to Supplementary Data File 1). Genes that are often mutated in PDAC were either not mutated in our or published ACC datasets, e.g., KRAS, or were rarely mutated in published (3 out of 26) or our datasets (2 out of 22), e.g., TP53[6, 7]. After employing MutSigCV[17], which identifies genes that are significantly mutated, no gene remained significantly enriched. Overall, no recurrent point mutations were identified in ACC suggesting that point mutations play a minor role in ACC.

**Mutational signatures identify risk factors for ACC.** To identify potential factors for tumor development, we calculated mutational signatures using the WES data. Different carcinogens such as UV light or tobacco have specific impacts on the mutational patterns of cancer[17]. Calculation of the mutational frequency of point mutations (i.e., C>A, C>G, C>T, T>A, T>C, T>G) in the context of one base upstream and one base downstream identified frequent C>A and C>T point mutations in ACC (Fig. 1d), which is similar to observations in other tumor entities (Supplementary Fig. 1a). Comparing these signatures to published mutational signatures as depicted in the COSMIC database using the deconstructSigs package[18] (Fig. 1e; Supplementary Fig. 1b), 18 out of the previously published 30 signatures were identified in ACC. Each tumor displayed on average 4 signatures (range: 1–6; for details refer to Supplementary Data File 2). Signature 1, corresponding to deamination of 5-methylcytosine is found in most cancer types. This mutational signature was found in 20 out of 22 ACC. Signature 4, which is related to tobacco smoking and was not observed in PDAC[19], was present in 12 tumors. In combination with tobacco chewing (signature 29), which was detectable in two patients, almost two-third (14 of 22) of ACC had mutational signatures associated with tobacco-associated carcinogens. Signatures associated with defective DNA repair (signatures 3, 6, 15, and 20) were identified in 15 of 22 tumors. Signature 3 is specifically associated with BRCA1 and BRCA2 mutations, whereas the other three are associated with defective DNA mismatch repair. Thus, ACC revealed distinct mutational signatures. This suggests defects in DNA double-strand break and DNA mismatch repair, which may contribute to the high genomic instability observed in this disease (see below).

**DNA methylation profiles suggest cell of origin.** To obtain a global picture of DNA methylation patterns and to identify copy number aberrations (CNA), we subjected 22 primary tumors, 12 metastases and 20 normal tissues from cohort 1 and 19 primary tumors and 10 normal tissues from cohort 2 to 450K analysis (for an overview refer to Supplementary Table 1). First, we assessed the similarity and quality of the two cohorts. (i) Cluster analysis of the two cohorts revealed that their global methylation pattern is very similar and comparable (Supplementary Fig. 2d). (ii) Very high tumor content achieved by microdissection by a pathologist was validated via the tumor purity estimation package LUMP[20] in 80 out of 83 samples (Supplementary Methods, Supplementary Fig. 2c). Principal component analysis (PCA) of all CpG sites of both cohorts revealed that normal pancreatic tissues clustered very closely together, whereas the tumors were distinct from the normal tissues and formed a wide-spread cluster, presenting a different global methylation pattern (Fig. 2a). To compare ACC to other pancreatic tumors, we further generated 450K data for 17 pancreatic neuroendocrine tumors (PNET) and used data for 146 PDAC generated by The Cancer Genome Atlas (TCGA)[21]. A DNA methylation phylogenetic tree separated the three

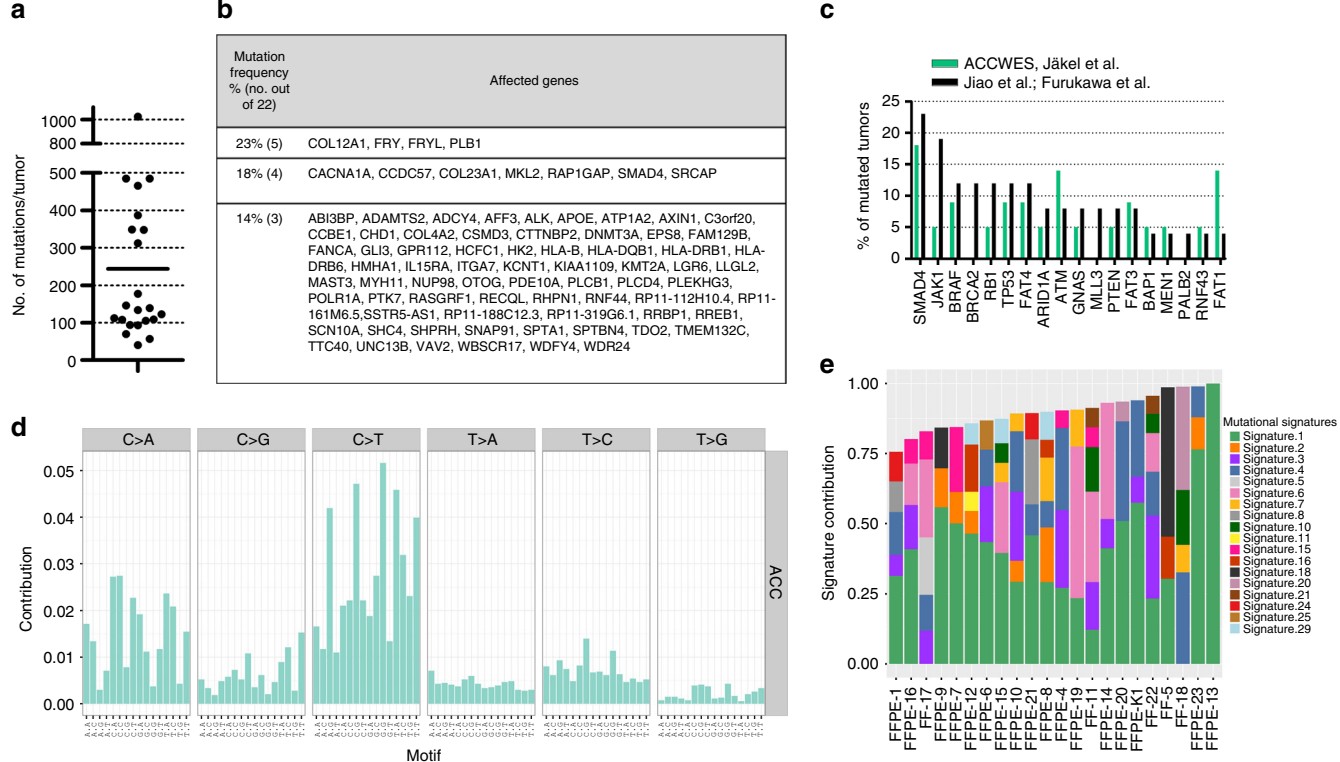

**Fig. 1** ACC display a high-mutational load and specific mutational signatures. **a** Number of mutated genes per tumor detected by WES. **b** Genes with most recurrent, however, still low frequency point mutations in ACC. **c** Comparison of WES results to published studies[12, 13] of ACC shows a similar low frequency of recurrent mutations. Both published studies[12, 13] were utilized as a comparison in combination as one of them only contained three ACC. **d** Mutational pattern of ACC. The type of base substitution in their genomic context (base preceding and base following) is displayed. **e** Published mutational signatures[27] present in each sample is displayed. Associations of signatures: signature 1: deamination of 5-methylcytosine, signature 4: tobacco smoking, signature 29: tobacco chewing, signature 3: DNA double-strand break repair defects, signature 6: defective DNA mismatch repair and microsatellite unstable, signature 15: defective DNA mismatch repair, signature 20: defective DNA mismatch repair

included pancreatic cancer types based on their methylome (Fig. 2b). As the cell of origin is unknown for ACC, we employed the algorithm of Houseman et al.[22] to infer the contribution of DNA methylation profiles of sorted normal pancreatic cell types (acinar, duct, neuroendocrine α-cells, and β-cells) to the methylation profiles of the pancreatic tumor types. The analysis revealed high similarities between the methylation profiles of (i) ACC and acinar cells (ii) PNET and α-cells, and (iii) PDAC and duct cells but also to some extent acinar cells (Supplementary Fig. 2a). Therefore, ACC should ideally be compared to healthy acinar cells for identifying differentially methylated regions. As formalin-fixed, paraffin-embedded (FFPE) tissues were used for 450K analysis and sorted acinar cells from adjacent tissue were not available, we used adjacent pancreatic tissue rich in acinar cells (~90% of cells) as reference for comparing tumors versus normal tissues.

We detected frequent and recurrent hypermethylation at 44,193 CpG sites in cohort 1 (Fig. 2c) and 26,959 sites in cohort 2 (Supplementary Fig. 2b). After mapping the CpG sites to either promoter sites, CpG islands, the overlap of CpG islands and promoters (CpG island promoters) or gene bodies (for details refer to "Methods"), we obtained differentially methylated regions in cohort 1 in a total number of 466, 690, 245, and 608 regions, respectively. Between 44 and 77% of these regions were validated in cohort 2, resulting in a final number of 245, 411, 189, and 270 regions differentially methylated in both cohorts (Fig. 2d, e, for beta values and differential methylation calling refer to Supplementary Data Files 3–5). Hypermethylation was predominant in these regions and only about one quarter of differentially

methylated regions in promoter CpG islands and genes were hypomethylated (Fig. 2d). A total number of 512 genes with differential methylation at their promoter site and/or gene body were identified (Supplementary Data File 6) in cohort 1 and validated in cohort 2. Gene ontology enrichment analysis of these differentially methylated regions was performed and revealed enrichment of developmental and cell adhesion pathways (Fig. 2f). The homophilic cell adhesion pathway including the protocadherins was the most enriched pathway (Supplementary Fig. 2e). Protocadherins are clustered in a 700 kb genomic region and hypermethylation of six promoters in this region was confirmed via MassARRAY (Supplementary Methods, Supplementary Fig. 2f).

Next, we investigated whether methylation changes upon metastases formation can be detected. Primary tumors always clustered close to their metastases (Supplementary Fig. 2c) and paired differential methylation analysis revealed only few differences, indicating that the global methylation profile remained stable in metastases (Supplementary Fig. 2d, e).

In summary, we found that (i) the methylation profile of ACC corresponds to non-neoplastic acinar cells, (ii) ACC display considerable differential methylation, and (iii) methylation aberrations of ACC metastases are comparable to their primary tumors.

**ACC harbor many chromosomal gains and losses**. 450K array signal intensities can be used to generate CNA maps[23]. Strikingly, ACC harbored many deletions and amplifications (Fig. 3a, cohort

1; Supplementary Fig. 3a, cohort 2), in particular a broad range loss of chromosome arm 1p (especially 1p36) and gain of chromosome 1q (especially 1q42). Localized highly recurrent deletions in 9p21.3, 16p13.3, and 18q21.2 loci and amplifications in 7p22.3 were also identified. In total, 2324 and 323 genes mapped to the deleted and amplified regions, respectively (only taking regions into account where not the whole chromosome arm was altered; Supplementary Data Files 7 and 8). About 62% of deleted and 11% of amplified genes were confirmed with the 450K data of cohort 2, resulting in 1441 and 35 confirmed deleted and amplified genes (Fig. 3b, c; Supplementary Data File 7). Four of these regions were additionally confirmed by qPCR (Supplementary Methods, Supplementary Fig. 5b). Thus, these large numbers of recurrent CNA were indicative of a high chromosomal instability in ACC. This instability did not increase when primary tumors were compared with their corresponding metastases (Supplementary Fig. 3b), similar to observations in PDAC[24].

**Integrative analysis identifies frequently altered genes**. We next integrated our data on DNA methylation and CNA. Circos plots showed that the hypermethylated and hypomethylated regions were quite evenly distributed throughout the genome, whereas the amplified and deleted regions tended to be localized (Fig. 4a; Supplementary Fig. 4). Each gene in each tumor was subsequently put into one of nine categories depending on whether the gene in that tumor was hypomethylated or hypermethylated at its promoter region, and/or amplified or deleted, or unaltered. This integrative approach revealed that previously not reported, frequent aberrations could be identified in ACC of which the 100 most affected genes are depicted in Fig. 4b. We then classified these aberrations using published reference lists[25–28] into the previously defined categories from these publications (known cancer genes, candidate cancer genes, and oncomirs from King's college[28]; tumor suppressors from Vanderbuilt;[27] mutated driver genes, driver genes CNA, and cancer predisposition genes from Vogelstein et al.;[26] epigenetic regulators from Plass et al.[25]). Using this approach, we discovered 292 genes significantly altered in ACC, which are already known to play a role in tumorigenesis in other cancer entities (Supplementary Data File 9).

**Specific tumor suppressors are frequently lost in ACC**. We next investigated whether the observed epigenetic and genetic aberrations in ACC are associated with a change of the respective protein expression. Immunohistochemical stainings were performed on a tissue microarray including 23 ACC from cohort 1 and 39 ACC from cohort 2 as well as 8 normal pancreatic samples. The protein expression of 8 aberrant genes was evaluated: *ARID1A*, *APC*, *CDKN2A*, *HIST1H*, *ID3*, *JAK1*, *PCDHG*, and *SOX2*. This selection of candidate genes from our integrative analysis results list was necessary for practical reasons such as experimental capacity and availability of high quality antibodies. There are however more potential cancer driver genes in our list, which could not be further evaluated within the presented study.

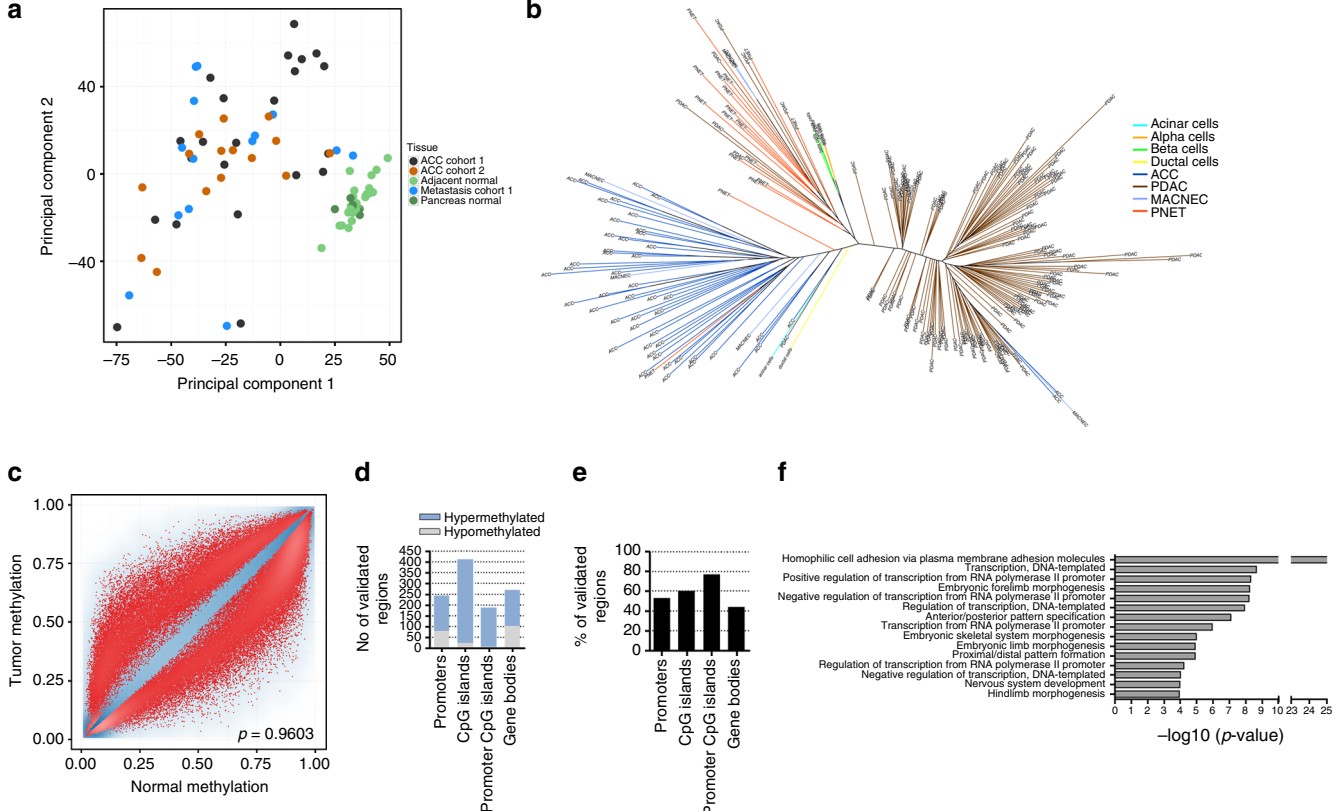

**Fig. 2** ACC display a unique methylome and many recurrent differentially methylated regions. **a** Principal component analysis of all CpG sites of cohort 1 and cohort 2. **b** DNA methylation phylogenetic tree, including ACC, MACNEC, PNET, PDAC, and sorted pancreatic cells. **c** Methylation scatterplot of all CpG sites on the 450K array in cohort 1 reveal massive differential methylation comparing all primary tumors versus all normal tissues. (red: sites above the rank cutoff, i.e., considered significant, blue: transparency corresponds to density of sites.). **d** Differential methylation defined by the rank cutoff in regulatory regions identified in cohort 1 and validated in cohort 2. **e** Validation rate of regions identified in cohort 1 and validated in cohort 2. **f** GO enrichment analysis displaying enriched pathways of the differentially methylated regions (input: regions of cohort 1 confirmed in cohort 2)

Four of these genes (*ARID1A*, *APC*, *CDKN2A*, and *ID3*) showed significantly reduced protein expression in the majority of tumor samples (Fig. 5a), whereas the remaining four genes displayed changes only in a few tumors (Supplementary Fig. 5a). ID3 was downregulated in 89 and 94%, ARID1A in 68 and 74%, APC in 71 and 62%, and CDKN2A in 53 and 52% of samples from cohort 1 and 2, respectively (Fig. 5b). Strikingly, most tumors displayed protein alterations in more than one of these four tumor suppressor genes (Fig. 5c). Nineteen tumors had alterations in all 4 genes, 21 tumors in 3 genes, and 13 tumors in 2 genes (Fig. 5d). Interestingly, ID3 and ARID1A alterations were evident in 60 out of 61 tumors, suggesting that downregulation of these two tumors suppressors are highly important. It was possible to predict the majority of protein losses based on the aberrant DNA methylation and CNA (Fig. 5e). Molecular alterations were additionally confirmed with independent methods (Supplementary Fig. 5b). As expected from the WES results, point mutations did not add any additional value to this. Taken together, we identified losses of several tumor suppressor genes which affect a majority of ACC suggesting an important role of these genes during the initiation and progression of these tumors.

## Discussion

In the present study, we investigated molecular aberrations in ACC on a genome-wide scale and on multiple levels. In recent years, ACC which represent a distinct tumor entity, that morphologically and immunohistochemically differ from PDAC or PNET, were shown to typically lack driver mutations known from the more common ductal carcinogenesis[6–8, 29]. We confirm that ACC do not display highly recurrent point mutations. However, our results show that ACC have highly unstable tumor genomes with localized, but also broad-ranged, chromosomal gains and losses. In addition, ACC exhibit mutational signatures that are associated with tobacco consumption and defective DNA repair. Common target genes clearly differ between ACC and other pancreatic tumors. For instance, ACC, but not PDAC or PNET harbor frequent aberrations in APC (60–70% versus <20%)[8, 30, 31], and exhibit a loss of ID3, which in contrast appears to be overexpressed in PDAC[32, 33]. Previously it has been shown that the tumors' methylation profiles resemble that of their cell of origin[34–36], thus one has to be careful when calculating differential methylation to actually identify cancer-related methylation changes. To that end, we used methylation profiles from sorted pancreatic cells. Morphologically, ACC resemble non-neoplastic

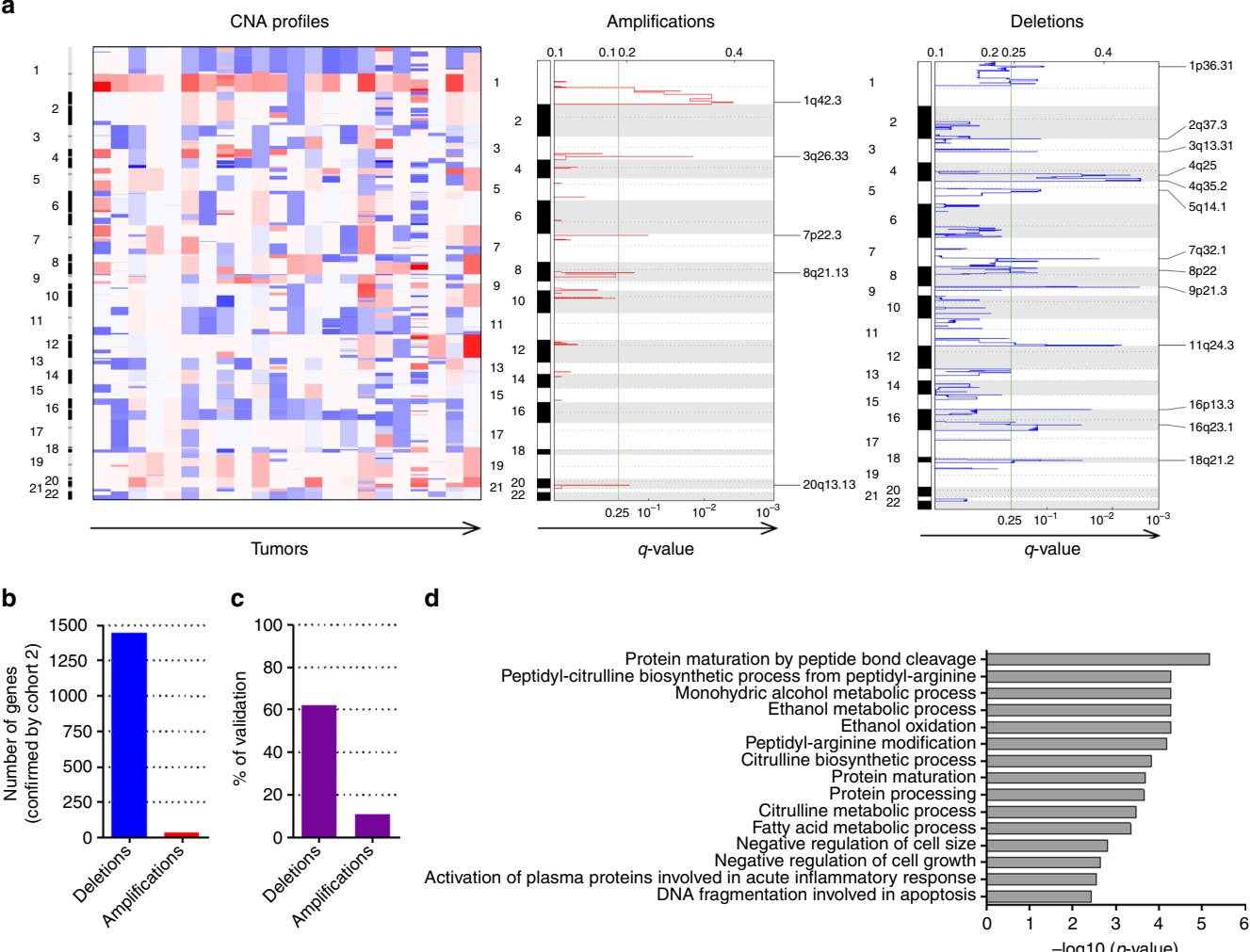

**Fig. 3** ACC are characterized by numerous copy number alterations, mainly deletions. **a** CNA profiles: heatmap of copy numbers calculated via the intensities of the 450K array (each tumor versus average normal) are displayed for each tumor of cohort 1 and each chromosome (red: amplifications, blue: deletions). Amplifications: *q* values of amplifications of all tumors of cohort 1. Deletions: *q* values of deletions of all tumors of cohort 1. **b** Confirmed number of genes mapping to significantly amplified or deleted regions. **c** Percentage of genes significantly amplified or deleted of cohort 1 that were confirmed in cohort 2. **d** GO enrichment analysis displaying enriched pathways of amplified and deleted genes (input: regions of cohort 1 confirmed in cohort 2)

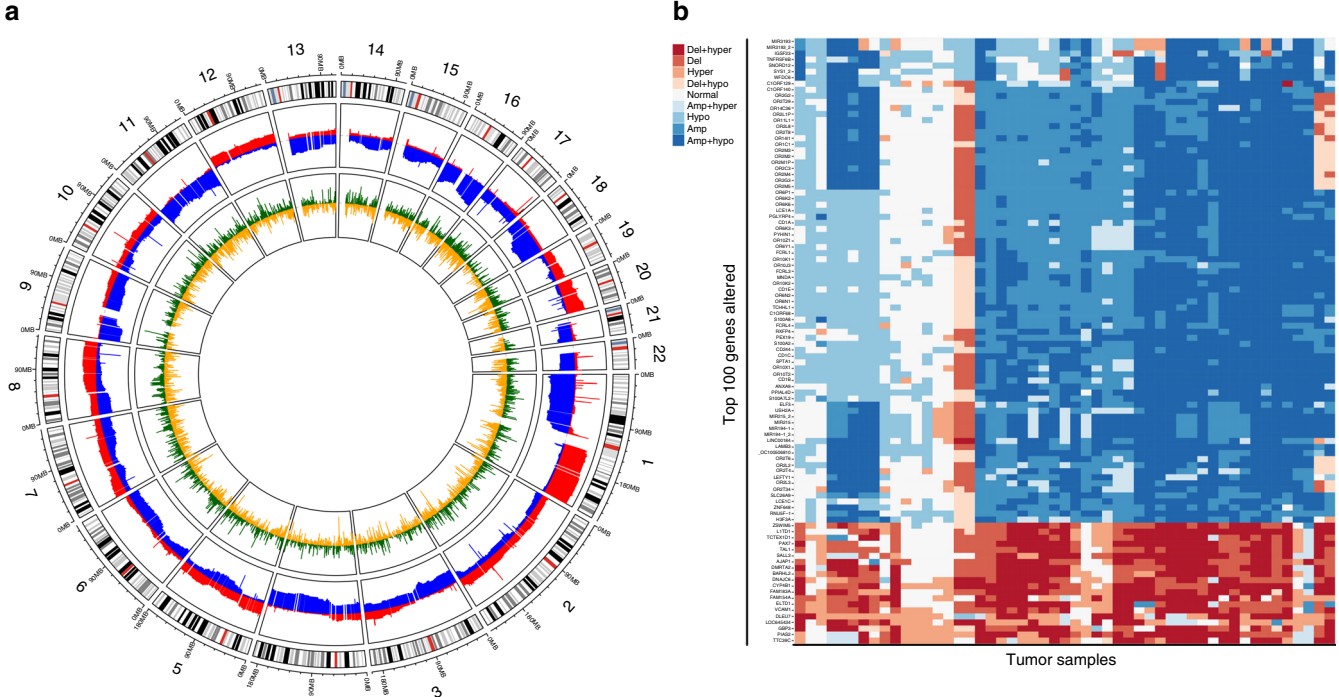

**Fig. 4** Frequent molecular alterations can be identified in ACC. **a** Circos plot displaying all autosomes (outer circle, grey, centrosomes red) with amplifications (middle circle, red), deletions (middle circle, blue), hypermethylation (inner circle, green), hypomethylation (inner circle, yellow). **b** Heatmap of top 100 genes encompassing copy number aberrations and differential promoter methylation; red: leading to loss of gene function, blue: leading to gain of gene function. The darker the color the more aberrations were detected

acinar cells[37], and this relation could also be reproduced on the molecular level, as ACC revealed a similar methylation profile to acinar cells. A further finding based on the methylation profiles of previously published sorted blood cells[22] is that ACC contain only few infiltrating immune cells, consisting mainly of CD8+ T-cells, which still has to be confirmed via immunohistochemistry (IHC). In contrast to that, PDAC are infiltrated by many immune cells with CD4+ T-cells prevailing (Supplementary Fig. 2a), as previously shown via IHC[38]. The molecular profile of MACNEC did not differ from pure ACC, suggesting a common molecular background despite morphological differences. However, we identified intertumoral differences based on the molecular aberrations in ACC. We were not able to explain these subgroups by clinical or molecular parameters, though (Supplementary Fig. 6). All these data further support the notion that these pancreatic cancers are distinct not only on the clinical and pathological but also profoundly different on the molecular level.

We used 450K arrays for copy number analysis, which enables its evaluation in parallel to DNA methylation analysis from the same DNA specimen[23]. It has been proven to be as robust and sensitive as SNP arrays. Although it may be less adequate than SNP arrays to detect alterations in large intergenic regions or gene desert regions, Feber et al.[39] showed that it provides good coverage of the majority of coding loci. ACC are characterized by an imbalanced genome with numerous chromosomal gains and losses (Fig. 3). This chromosomal instability may be explained by (i) loss of ARID1A, BRCA1/2[13], and CENPE (Supplementary Data File 9), and (ii) mutational signatures associated with defective DNA repair mechanisms. ARID1A protein is lost in ~70% of the tumors. As part of the SWI/SNF complex, it is involved in non-homologous end joining[40], maintains proper chromosome segregation and prevents anaphase bridges during mitosis[41]. Its loss might contribute to the massive chromosomal

gains and losses observed in ACC. BRCA2 is also important in maintaining genomic stability because of its involvement in homologous recombination. Furukawa et al.[13] reported BRCA2 point mutation in 3 out of 7 ACC and a downregulation at the protein level in 5 out of 11 ACC. Others have reported lower BRCA2 mutation frequencies[2]. In our series, about 70% of the ACCs had mutational signatures associated with DNA repair defects. The mutational signature 3, which is thought to be caused by BRCA1 and/or BRCA2 loss, can be detected in 10 out of the 22 sequenced tumors. Signatures that are associated with DNA mismatch repair occured in 11 tumors. These data reflect the important role that defective DNA repair pathways may have in ACC tumorigenesis.

In addition, cell cycle control is impaired in ACC by frequent abrogation of APC, CDKN2A, and ID3, which are all negative regulators of the cell cycle (Fig. 6). APC inhibits CTNNB1, which in turn activates the transcription factors (TCFs). One of the target genes of TCFs is cyclin D (CCND), which is a regulator of cell cycle progression[42, 43]. In ACC, the cell cycle inhibitor CDKN2A was so far not reported to be frequently affected[12, 44], mostly due to the fact that in these studies only mutations had been investigated. Here, we report an absence of CDKN2A protein expression in >50% of cases and an overexpression in 8 cases suggesting alterations within the pRb pathway[45]. Nearly, all tumors (~90%) exhibit a decrease in ID3 protein expression. ID3 inhibits TCFs by forming non-functional heterodimers and thereby prevents transcription of the target genes, e.g., TCF3, which in turn activates cyclin D3 and cyclin E[32, 46]. TCF3 further regulates acinar cell identity by activating expression of the pancreatic transcription factor PTF1A[33] and acinar transcription factor BHLHA15 (=Mist-1)[47]. A loss of ID3, therefore should lead to an overexpression of TCF3, resulting in an activation of the acinar cell program, whereas an activation of ID3 as shown by

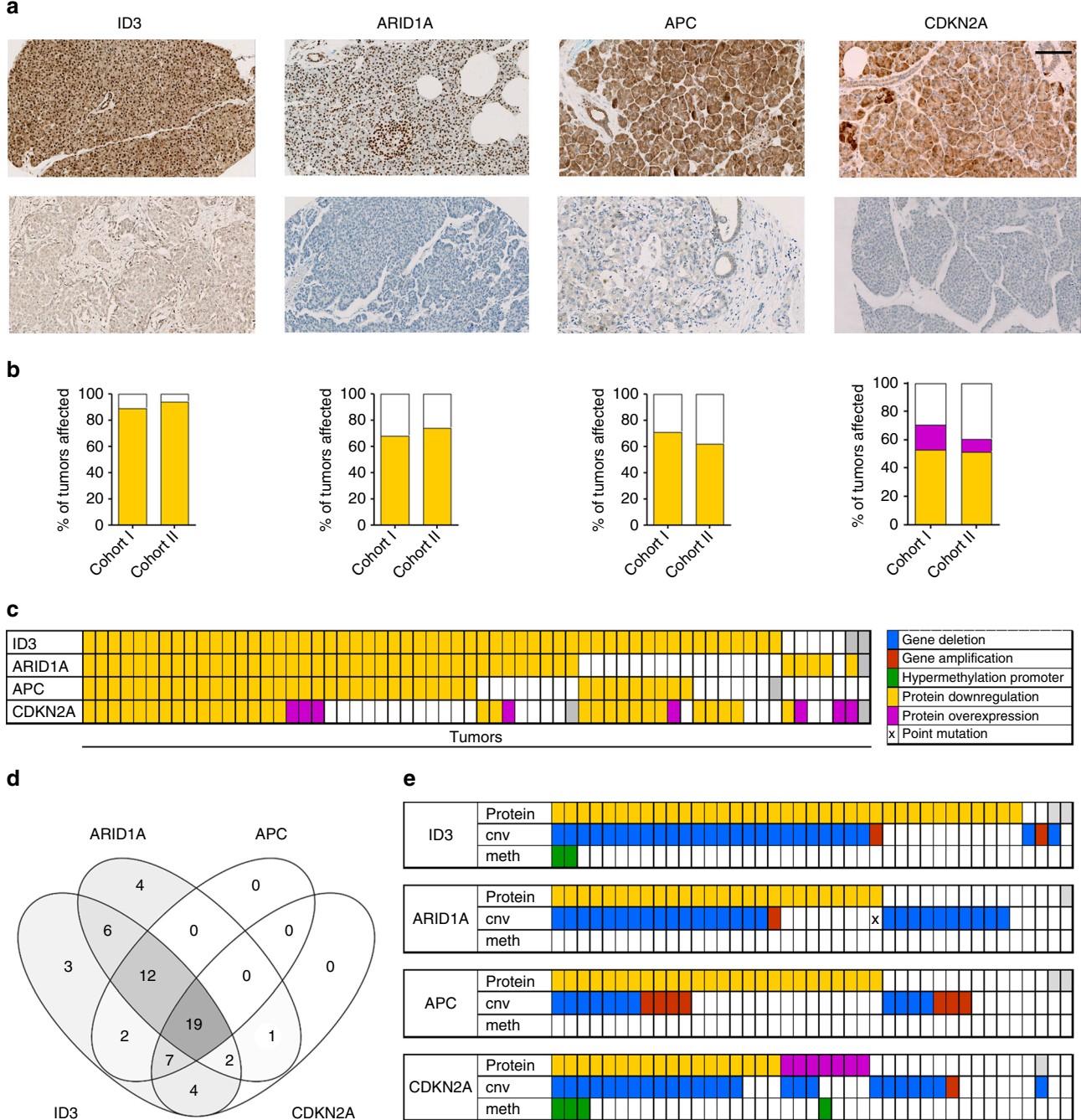

**Fig. 5** ID3, ARID1A, APC, and CDKN2A are downregulated in the vast majority of ACC. **a** representative IHC for tumor and normal tissue; scale bar corresponds to 100 μm; all microscopy images are shown at the same magnification. **b** Percentages of altered tumors in cohort 1 and cohort 2, a total of 69 tumors were investigated. **c** Overview of co-occurrence of protein alterations. **d** Venn diagram of co-occurrence of protein alterations. **e** IHC in combination with 450K data: co-occurrence of (epi-)genetic and protein alterations, grey: no data available

Kim et al.[47] leads to an inactivation of the acinar cell program in PDAC. This gives rise to the hypothesis that ID3 expression might be the switch that distinguishes PDAC from ACC.

Our molecular characterization of ACC offers therapeutic targets for drugs currently tested in clinical trials for other tumors entities or that are already EMA-approved and/or FDA-approved (Supplementary Data File 10). This opens the opportunity that ACC patients can be included in basket trials, in which not the organ of the cancer but the molecular alterations are defining the inclusion criteria[48]. For instance, alterations in CDKN2A, ID3, and APC can be exploited by molecules interfering with the cell

cycle regulator, CDK4/6[49]. APC-altered tumors might additionally benefit from inhibiting one of the Wnt pathway members[50]. And in tumors lacking ARID1A, PI3K as well as ARID1B, EZH2 and PARP inhibition might lead to effective therapies[51–55] (Supplementary Data File 10).

In summary, our study revealed that ACC are characterized by numerous copy number alterations and aberrantly methylated sites, and display distinct mutational signatures. Our analysis identified the four tumor suppressor genes ID3, ARID1A, APC, and CDKN2A as recurrently affected in ACC. The latter three have been reported as driver

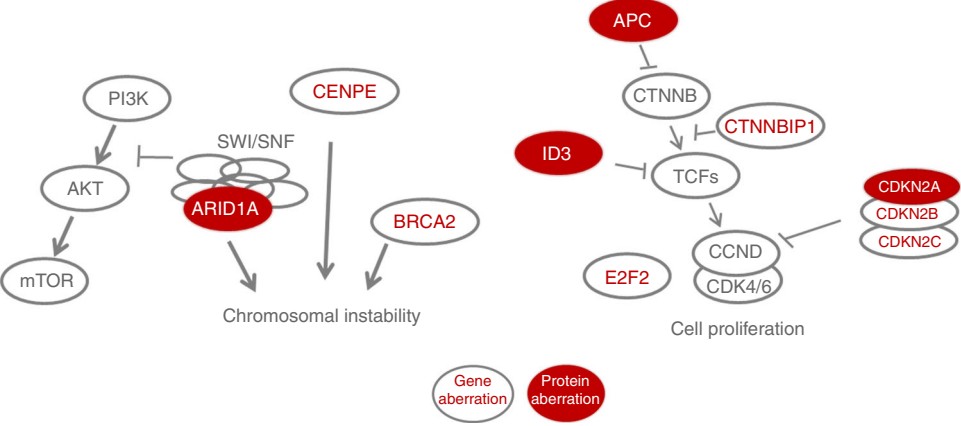

**Fig. 6** Analyses reveal aberrations in genome stability and cell cycle control in ACC

genes in tumorigenesis[26], therefore aberrations of these genes have to be considered for the development and progression of this disease. Moreover, these findings might offer insights for the development of treatment options for ACC.

## Methods

**Study cohorts**. Two independent ACC cohorts were investigated (for details refer to Supplementary Table 1). Cohort 1 is a collection of 34 tumors (22 primary tumors and 12 metastases) and 24 normal pancreatic tissues, which were obtained from the Institute of Pathology, University of Heidelberg, Germany. Primary tumors from this cohort were used as discovery cohort, as matched germline controls were available for sequencing. Study cohort 2 was used as validation cohort and consists of 39 tumors (38 primary tumors and 1 metastasis) and 10 normal pancreatic tissues, which were obtained from the Consultation Center for Pancreatic and Endocrine Tumors of the Department of Pathology, Technical University Munich, Germany. Twenty and thirty-eight primary tumors and 3 and 1 metastases from cohort 1 and 2, respectively, and 8 normal pancreatic samples were further combined on a tissue microarray to facilitate immunohistochemical stainings. Most tissues were available as formalin fixed and paraffin embedded (FFPE) tissues, for five tumors fresh-frozen material was available. In addition, DNA from 17 pancreatic PNET obtained from FFPE tissue were investigated. Tumors were microdissected by a pathologist with a tumor purity of >90%. Tissue samples were provided by the tissue bank of the National Center for Tumor Diseases (NCT, Heidelberg, Germany) in accordance with the regulations of the tissue bank and the approval of the ethics committee of Heidelberg University (No. 207/2005; informed consent was obtained from all participants). The results shown on PDAC are based upon data generated by the TCGA Research Network: http://cancergenome.nih.gov.

**Conduct of Infinium HumanMethylation450 BeadChip array**. DNA was isolated from FFPE tissue using the Invisorb Genomic DNA Kit II (Stratek, Birkenfeld, Germany). Quality was evaluated by real-time PCR analysis on Light Cycler 480 Real-Time PCR System (Roche, Mannheim, Germany) using the Infinium HD FFPE QC Kit (Illumina, San Diego, USA). DNA samples were bisulfite treated with the EZ-96 DNA Methylation Kit (Zymo Research Corporation, Orange, USA) and subsequently treated with the Infinium HD DNA Restoration Kit (Illumina). Each sample was whole genome amplified and enzymatically fragmented following the instructions in the Infinium HD FFPE Methylation Guide. The DNA was applied to the 450K array (Illumina) and hybridization was performed for 16–24 h at 48 °C. Microarray scanning was done using an iScan array scanner (Illumina).

**Analysis of genome-wide DNA methylation via 450K**. Analysis was run with the R Package RnBeads[56]. In summary, data were subjected to quality control, normalization with BMIQ[57], and differential methylation was then called between groups using RnBeads's rank cutoff, which implements the difference in mean methylation, the quotient in mean methylation and the *p*-value obtained by limma test (for multiple comparisons) or Student's *t*-test (for paired analysis). Region annotations were implemented to analyze the average differential methylation in the following regions: gene bodies (defined by Ensembl[58]), promoter regions defined by RnBeads[56] (regions spanning 1.5 kilo base pairs (kb) upstream and 0.5 kb downstream of the transcription start site for every gene[56]), CpG Islands (defined according to the definition of the UCSC browser[59]), and promoter segments (intersection of promoters and CpG islands). For investigation of recurring methylation changes from primary tumors to metastasis, paired *t*-test was used to calculate differences. For cell type contributions, the method published by

Houseman et al.[22] was used, which is a method similar to regression calibration. 450K data from normal hematopoietic cell types were obtained from Reinius et al.[60] In addition to the blood sample methylation data from this work, we added DNA methylation profiles from sorted pancreatic cells (acinar, duct, alpha, and beta cells). Phylogenetic trees were generated calculating Euclidean distance matrices based on all CpG sites on the array by the minimal evolution method[61] using the fastme.bal function from the R package ape[62].

**Analysis of genome-wide copy number via 450K**. The Bioconductor package conumee[63] was used to calculate copy number alterations from the intensities obtained from the 450K array (bin probe size was set to 5, rest of parameters set to default). Gistic[64] was then employed to investigate frequently deleted/amplified regions/genes (with default parameters).

**Microsatellite instability analysis**. Microsatellite instability was previously assessed in a subset of the here presented tumor samples in Bergmann et al.[15] (also refer to Supplementary Data File 1), and analyses were conducted using the marker panels CAT25, BAT25, and BAT26 as developed by Findeisen et al.[65] In brief, tumors were classified as high-level microsatellite instable (MSI), if at least two of the three markers displayed instability. PCR primers for the amplification of CAT25 were, forward 5V-CCTAGAAACCTTTATCCCTGCTT-3V, and reverse 5V-GAGCTTGCAGTGAGCTGAGA-3V. PCR primers were labeled at the 5Vend with FITC (BAT26 and CAT25) or HEX fluorescent dye (BAT25), respectively. Multiplex PCR was carried out in a total reaction volume of 25 μl using a final concentration of 200 μmol l$^{-1}$ deoxynucleotide triphosphates, 12.5 pmol l$^{-1}$ of each primer, 1× PCR buffer (20 mmol l$^{-1}$ Tris-HCl (pH 8.4), 50 mmol l$^{-1}$ KCl), 1.5 mmol l$^{-1}$ MgCl$_2$, and 0.75 U of Taq DNA polymerase (Life Technologies/BRL, Eggenstein, Germany). Genomic DNA (50 ng) was used as a template. Reaction mixes were subjected to the following conditions: initial denaturation at 94 °C for 5 min followed by 38 cycles of denaturation at 94 °C for 30 s, annealing at 55 °C for 30 s, extension at 72 °C for 30 s, and a final extension step at 72 °C for 7 min.

**Whole-exome sequencing**. Fresh frozen samples were checked and selected with H&E stainings for high tumor cellularity. For DNA from fresh-frozen samples, library preparation was performed according to Agilent's "SureSelectXT Target Enrichment System for Illumina Paired-End Multiplexed Sequencing Library" kit, whereas for DNA from FFPE tissue the "SureSelect Automated Library Prep and Capture System SureSelectXT Automated Target Enrichment for Illumina Paired-End Multiplexed Sequencing" kit was used. Samples were run paired-end (125 bp) on a HiSeq 2000 v4. Alignment of data were processed by the following parameters: reference genome: hs37d5, alignment program: bwa-0.7.8 mem, alignment parameter: -T 0, duplication marking program: picard-1.125, default duplication marking program parameters were used (https://broadinstitute.github.io/picard/command-line-overview.html#MarkDuplicates). Alterations that are likely to be benign, so called FLAGS[66] were excluded. Previously published WES studies of ACC[13, 14] were utilized as a comparison to the here generated WES results. MutSigCV was used to calculate significantly recurrent mutations[67].

**Integrative analysis**. To integrate changes in methylation and in copy numbers within the genes, data were categorized into the following nine groups: deleted and promoter hypermethylated, deleted only, promoter hypermethylated only, deleted and promoter hypomethylated, unaltered, amplified and promoter hypermethylated, promoter hypomethylated only, amplified only, and amplified and promoter hypomethylated. The following cutoffs were used: for copy number alterations the deletion and amplification thresholds from Gistic were employed. For differential methylation, a tumor was called to be hypo-/hypermethylated at a gene if the

promoter methylation was below or above 20% of the mean methylation level of all normal pancreatic tissues.

**Identification of cancer-related aberrations**. Significantly differentially methylated genes and genes altered on the copy number levels were overlapped with multiple previously published gene lists in order to obtain a list as comprehensive as possible of cancer related genes (known cancer genes, candidate cancer genes, and oncomirs from King's college (http://ncg.kcl.ac.uk/download.php);[28] tumor suppressors from Vanderbuilt (https://bioinfo.uth.edu/TSGene/Human_TSGs.txt);[27] driver genes mutated, driver genes CNA, and cancer predisposition genes from Vogelstein et al.;[26] epigenetic regulators from Plass et al.[25]).

**Circos plots**. The R package circlize was used to create circos plots. For copy number aberrations the total number of tumors which are amplified (red) or deleted (blue) out of all 41 primary tumors that were subjected to 450K analysis is depicted. For methylation the difference of the promoter region means (tumor minus normal tissue) is depicted (hypermethylation dark green, hypomethylation yellow).

**GO enrichment**. For gene enrichment analysis the function annotation tool from the DAVID website was used[68].

**Immunohistochemical staining**. Immunohistochemical analyses were performed using an automated slide staining system (Ventana BenchMark Ultra, Roche) and the avidin–biotin complex method[15]. All information on antibodies, such as source, catalogue and lot numbers, concentrations etc., is listed in Supplementary Data File 11. The secondary antibody was integrated in the used staining kit (Ventana Optiview DAB; Roche). Pretreatment was performed using the buffer ULTRA Cell Conditioning Solution CC1 (Roche), as listed in Supplementary Data File 11. Staining intensities were evaluated by a pathologist. Proteins were considered downregulated when the value of the tumor was at least one staining intensity lower than the lowest staining intensity of the normal tissues, whereas proteins were considered upregulated when the value of the tumor was at least one staining intensity higher than the highest staining intensity of the normal tissues.

**Mutational signatures**. Mutational frequencies were plotted using the SomaticSignatures package (version 2.8.4) available on Bioconductor[69]. WES data from ACC and publicly available WES from TCGA (available in the SomaticCancerAlterations Bioconductor package 1.8.2) were used to assess frequency of point mutations in the context of three nucleotides (X-A/T-X)[70]. For investigating the occurrence of published mutational signatures summarized by COSMIC (http://cancer.sanger.ac.uk/cosmic/signatures) in ACC, we employed deconstructSigs (version 1.8.0)[18].

**Isolation of pure pancreatic cells**. Pancreatic cells were isolated by FACS with antibodies specific for duct, acinar, endocrine alpha, and endocrine beta cells[71]. In brief, pancreatic cells were dispersed using trypsin (10 min at 37 °C) and occasional mechanical disruption with a p1000 micropipette. Cells were strained (40 μm), washed, resuspended, and incubated with the appropriate 1:50 diluted primary antibodies (Alpha cells: Pa1 Antibody (DHIC2-2C12), pan-islet cells: HPi2 Antibody (HIC1-2B4.2B), acinar cells: HPx1 Antibody (HIC0-3B3), and duct cells: HPd1 Antibody (DHIC2-4A10) from Novus Biologicals (catalogue# NBP1-18949, NBP1-18946, NBP1-18951, and NBP1-18953). Secondary antibodies (PE-conjugated anti-mouse IgM (μ chain) and Dylight488-conjugated anti-mouse IgG [1 + 2a + 3]; Jackson ImmunoResearch, catalog# 115-545-164 and 115-116-075) were added at 1:200 and cells were subsequently sorted with an inFlux V-GS (BD Biosciences) equipped with a 100 μm nozzle. Duct and acinar cell 450K data from Lehmann-Werman et al.[36] was used and endocrine α-cell and β-cell 450K data were additionally generated.

**Data availability**. The methylation array and exome-sequencing data of all generated datasets are deposited at the European Genome-Phenome Archive under study accession number EGAS00001002533. Data for PDAC were downloaded from the TCGA portal (https://portal.gdc.cancer.gov/projects/TCGA-PAAD). All other relevant data are available from the authors.

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

## Acknowledgements

We thank the microarray unit and the high-throughput sequencing unit of the DKFZ Genomics and Proteomics Core Facility for providing the Illumina HumanMethylation 450 arrays, performing whole-exome sequencing and related services. We thank the Data Management and Genomics IT from the eilslabs for aligning sequencing data and calling SNVs. Funding for this study comes in part from the German Cancer Consortium (DKTK) and the Helmholtz Foundation.

## Author contributions

C.J., F.B., C.P., O.P., and P.S. designed and supervised the study. C.J., F.B., and D.v.d.D. performed experiments. C.J., R.T., and Y.A. analyzed data. F.B, O.S., T.H., G.K., and P.Schi. acquired and processed ACC patient tissues, acquired and annotated clinical and follow-up data. C.D., M.G., J.M., and Y.D. generated data for sorted pancreatic cells. C.J., O.P., and P.S. wrote the manuscript. F.B., R.T., C.P., O.P., and P.S. gave conceptual advice. All authors interpreted the results and implications and contributed to the final manuscript.

## Additional information

**Competing interests:** The authors declare no competing financial interests.

