## [Peer Review File · Nature Communications]

Reviewers' comments:

Reviewer #1 (Remarks to the Author):

The manuscript by Jakel and colleagues entitled "Genome-wide genetic and epigenetic analyses of pancreatic acinar cell carcinomas reveal aberrations in genome stability and cell cycle control" presents a multi-omic analysis of two cohorts of ACC samples, in total 41 primary tumours, 12 metastases and 30 normal tissues, plus other samples sets for comparison. Overall the study shows some novel findings including the lack of recurrent point mutations, a mutational signature that imply DNA repair pathways contribute significantly to carcinogenesis, and that tobacco smoking mutation patterns can be observed. These results also show that methylation profiles suggest the cell of origin is likely to be acinar cells, which is not unexpected. The study is well conducted and presents novel data that might help future studies. The limitations of the study are in the study size, which is relatively small compared to other tumour profiling studies and the reliance on 450k methylation data for copy number analysis (which provides relatively low resolution).

Further comments and minor corrections.

1. It would be helpful to know how rare this tumour type is in comparison to other pancreatic histologies, and how rare in relation to other cancer types. If it is a rare subtype of a rare cancer, then that explains the small numbers, and puts it in context.
2. Results "confirm" the previously reported lack of recurrent mutations, rather than reveal. It is important to make this distinction in the text of the manuscript
3. Tobacco association with mutation signatures is interesting. Is there any epidemiological evidence that supports a role for smoking as a risk factor in comparison to other subtypes that don't display this signature as prominently?
4. 450k methylation for copy number is relatively low resolution, and should not be referred to as "high resolution maps"
5. regarding the classification of tumours into 9 categories and the group of "hypo- or hyper-methylated". The location of the methylation is critical to the interpretation of this, this analysis should only focus on promoter methylation for it to be relevant to the rest of the categories.
6. Discussion – the word "instable" should be "unstable"
7. Discussion – should include a few statements on the limitations of this study
8. Supplementary figure 2, part A – it is not clear what this scale is, is it correlation?
9. Supplementary Figure 4 and 5 appear to be labelled in correctly. Figure 5 could include some statistical analysis of the groups if any differences are inferred from these data.

Reviewer #2 (Remarks to the Author):

In this study, Jakel and colleagues report comprehensive molecular characterization of a large series of pancreatic acinar cell carcinomas. Although whole exome sequencing of this rare tumor type has been previously reported, the submitted study represents the largest series to date, and it is the first to comprehensively assess changes in methylation and copy number in addition to single base substitutions and small indels. As such, it represents an important contribution to our understanding of pancreatic acinar cell carcinomas. Still, the study as written could be improved with attention to the following.

1. Previous reports have demonstrated that a subset of acinar cell carcinomas have microsatellite instability. The large number of mutations in some tumors in the current cohort raises this possibility. Can the authors assess microsatellite instability in their analyzed tumors, at least in the tumors with a large number of mutations? Are there any molecular alterations that could underlie this phenotype (i.e. mutations in DNA mismatch repair genes) in their cohort? Microsatellite instability has great clinical implications, as these tumors respond well to immunotherapy.
2. The authors report that there are no recurrent point mutations in ACCs, but then the provide a

list of genes that are mutated in >20% of analyzed tumors. Though not mutated in the majority of ACCs, these genes still have recurrent mutations and could be drivers. Still, this cannot be assessed from the data given. Can the authors report the mutation frequency after excluding any samples with an unusually large number of mutations (such as the one with >1000 mutations)? In addition, the authors should use an algorithm such as MutSigCV (or similar) to correct for gene size, nucleotide composition, etc to determine which genes are mutated more than expected by chance.

3. The 8 tumor suppressor genes were chosen for in depth analysis based on the availability of antibodies for IHC. While such arbitrary choices are necessary, the authors should acknowledge this in the Discussion. Although they provide good evidence for the importance of the 4 proteins highlighted, there may be many more equally important tumor suppressor genes that don't currently have good antibodies.

Point-by-point response letter for manuscript NCOMMS-17-05277-T

Jäkel et al. " *Genome-wide genetic and epigenetic analyses of pancreatic acinar cell carcinomas reveal aberrations in genome stability and cell cycle control* ".

We thank both referees and the editor of *Nature Communications* for giving us valuable advice and the opportunity to improve our manuscript.

Reply to reviewer #1

The manuscript by Jakel and colleagues entitled "Genome-wide genetic and epigenetic analyses of pancreatic acinar cell carcinomas reveal aberrations in genome stability and cell cycle control" presents a multi-omic analysis of two cohorts of ACC samples, in total 41 primary tumors, 12 metastases and 30 normal tissues, plus other samples sets for comparison. Overall the study shows some novel findings including the lack of recurrent point mutations, a mutational signature that imply DNA repair pathways contribute significantly to carcinogenesis, and that tobacco smoking mutation patterns can be observed. These results also show that methylation profiles suggest the cell of origin is likely to be acinar cells, which is not unexpected. The study is well conducted and presents novel data that might help future studies. The limitations of the study are in the study size, which is relatively small compared to other tumor profiling studies and the reliance on 450k methylation data for copy number analysis (which provides relatively low resolution). Further comments and minor corrections.

1. It would be helpful to know how rare this tumor type is in comparison to other pancreatic histologies, and how rare in relation to other cancer types. If it is a rare subtype of a rare cancer, then that explains the small numbers, and puts it in context.

We agree with the reviewer and added the information (page 2, introduction section, lines 34-36) that ACC is a very rare pancreatic neoplasm that accounts for only less than 2% of all pancreatic cancers. This illustrates that our sample collection of 73 ACC tissues represents one of the largest available tissue-based collectives world-wide.

2. Results "confirm" the previously reported lack of recurrent mutations, rather than reveal. It is important to make this distinction in the text of the manuscript.

In lines 58 and 62, the term "reveal" is replaced by the word "confirm" to specify that the observed lack of recurrent mutations had previously been reported.

3. Tobacco association with mutation signatures is interesting. Is there any epidemiological evidence that supports a role for smoking as a risk factor in comparison to other subtypes that don't display this signature as prominently?

Smoking is regarded as a known risk factor for pancreatic cancer (Raimondi et al. *Nat Rev Gastroenterol Hepatol* 6, 699, 2009). The published data relate however mainly to pancreatic ductal adenocarcinoma (PDAC) constituting ca. 90% of all pancreatic cancers. For rare pancreatic cancer subtypes such as ACC, sufficient epidemiological data are not available to draw adequate conclusions. Nevertheless, a recent analysis of Alexandrov et al. (*Science* 354(6312):618-622, 2016) investigated mutational signatures associated with tobacco smoking in 17 human cancer types

including PDAC. Overall, their results reveal that tobacco-associated signature 4 mutations occurred more often in cancers from smokers compared with nonsmokers. They observed this signature however not in PDAC (we added this to the main text; line 89), whereas in our study, more than 54% of ACCs showed signature 4 mutations. This again illustrates that PDAC and ACC are pancreatic cancer subtypes which carry different mutational signatures pointing to different cancer etiologies. Further elucidation of the association of signature 4 mutations and smoking in ACC in future studies is warranted but very challenging, as epidemiological data of large cohorts of this rare pancreatic cancer subtype will be required.

4. 450k methylation for copy number is relatively low resolution, and should not be referred to as “high resolution maps”.

We changed this sentence according to the reviewer’s comment (lines 149-150)

5. Regarding the classification of tumors into 9 categories and the group of “hypo- or hyper-methylated”. The location of the methylation is critical to the interpretation of this, this analysis should only focus on promoter methylation for it to be relevant to the rest of the categories.

We thank the reviewer for pointing this out. The analysis was indeed performed using promoter regions. We apologize for the incorrect labeling. We added this information to the main text (line 168), supplementary file (lines 79-80) and in the legend of Figure 4.

6. Discussion – the word “instable” should be “unstable”

We changed this sentence according to the reviewer’s comment (line 223)

7. Discussion – should include a few statements on the limitations of this study.

The reviewer addresses two possible limitations of our study: (i) the relatively small number of ACC samples, and (ii) the reliance on 450k methylation data for copy number analysis.

Ad (i); the information that ACC is a very rare pancreatic neoplasm accounting for less than 2% of all pancreatic cancers was missing to the reviewer (see point 1.). We have therefore added it to the text, and we think in the case of this rare cancer subtype that 73 ACC tissue samples represent a substantial collection being one of the largest available tissue-based collectives world-wide.

Ad (ii); for copy number analysis, we used a method developed by Stephan Beck’s lab, University College London, which enables the evaluation of both DNA methylation and CNA from the same DNA specimen (Morris et al. 2014). They compared copy number data from 450k and SNP arrays (Feber et al. 2014) and showed that the 450k arrays are as robust and sensitive as current high density SNP arrays for the detection of CNA. With regard to resolution, Feber et al. concluded from their data that 450 arrays “may lack the resolution of SNP arrays to detect alterations in large intergenic regions or gene desert regions, they provide however high resolution coverage of the majority of coding loci”. We have included this point to the Discussion section (lines 244-248).

8. Supplementary Figure 2, part a – it is not clear what this scale is, is it correlation?

We thank the reviewer for pointing this out. We added the information to the supplementary file (lines 167-168).

9. Supplementary Figure 4 and 5 appear to be labelled incorrectly. Figure 5 could include some statistical analysis of the groups if any differences are inferred from these data.

We apologize for incorrect labeling and that a previous version of Supplementary Figure 5 was presented. We checked and corrected the figure and figure legend in the supplementary file (lines

200-212). We did not include group-wise comparisons as we did not infer any group differences from these data. For the definition of hypo- and hypermethylation, please refer to the method 'integrative analysis' in the supplementary file (lines 82-84).

Reply to Reviewer #2

In this study, Jakel and colleagues report comprehensive molecular characterization of a large series of pancreatic acinar cell carcinomas. Although whole exome sequencing of this rare tumor type has been previously reported, the submitted study represents the largest series to date, and it is the first to comprehensively assess changes in methylation and copy number in addition to single base substitutions and small indels. As such, it represents an important contribution to our understanding of pancreatic acinar cell carcinomas. Still, the study as written could be improved with attention to the following.

1. Previous reports have demonstrated that a subset of acinar cell carcinomas have microsatellite instability. The large number of mutations in some tumors in the current cohort raises this possibility. Can the authors assess microsatellite instability in their analyzed tumors, at least in the tumors with a large number of mutations? Are there any molecular alterations that could underlie this phenotype (i.e. mutations in DNA mismatch repair genes) in their cohort? Microsatellite instability has great clinical implications, as these tumors respond well to immunotherapy.

We thank the reviewer for this suggestion. The microsatellite instability (MSI)-status for some of our primary tumors samples has been previously determined (Bergman et. al, 2014). We used these data and revealed that the tumor with the highest mutational load (>1000 mutations) was MSI positive while the remaining investigated tumor samples were all MSI negative, i.e. microsatellite stable (MSS). Loss of protein expression as a surrogate marker for mutations in *MLH1*, *MSH2* and *MSH6* -which could explain the MSI phenotype - were not detected, and exome sequencing data did also not reveal mutations in these genes. We added this information to Supplementary Table 2, the online methods (lines 57-60) and in the main text (lines 67-68).

2. The authors report that there are no recurrent point mutations in ACCs, but then they provide a list of genes that are mutated in >20% of analyzed tumors. Though not mutated in the majority of ACCs, these genes still have recurrent mutations and could be drivers. Still, this cannot be assessed from the data given. Can the authors report the mutation frequency after excluding any samples with an unusually large number of mutations (such as the one with >1000 mutations)? In addition, the authors should use an algorithm such as MutSigCV (or similar) to correct for gene size, nucleotide composition, etc. to determine which genes are mutated more than expected by chance.

We thank the reviewer for this suggestion. We applied the MutSigCV algorithm (Lawrence et al. Nature 2013) to our ACC exome sequences, and observed that no gene remained significantly recurrent after this analysis. We added this information to the manuscript (lines 74-75) and the supplementary methods (lines 74-75).

3. The 8 tumor suppressor genes were chosen for in depth analysis based on the availability of antibodies for IHC. While such arbitrary choices are necessary, the authors should acknowledge this in the Discussion. Although they provide good evidence for the importance of the 4 proteins highlighted, there may be many more equally important tumor suppressor genes that don't

currently have good antibodies.

Our integrative analysis identified a list of frequently altered genes in ACC. We selected eight aberrant genes from the top candidates of this list to subsequently investigate their protein expression. For practical reasons, the selection was based on experimental capacity and the availability of high quality antibodies enabling an adequate evaluation of immunohistochemically stainings on the ACC tissue microarrays. We are however aware that our gene list (supplementary table 9) probably contains other potential cancer driver genes, which could not be further evaluated within the presented study. As suggested by the reviewer, we added this information to the manuscript (lines 184-188).

REVIEWERS' COMMENTS:

Reviewer #1 (Remarks to the Author):

Overall, the authors have addressed the comments made by the reviewers. There is one potential error I noticed in the supplementary figures with the circos plots for Chr1 and Chr2 potentially mixed up. There is a large loss/gain for Chr 1 in the main Figure, but this appears on chr2 in the supplementary figure, which is potentially mislabelled.

Reviewer #2 (Remarks to the Author):

The authors have appropriately addressed my concerns in the revised manuscript.

Point-by-point response letter for manuscript NCOMMS-17-05277A

Jäkel et al. "*Genome-wide genetic and epigenetic analyses of pancreatic acinar cell carcinomas reveal aberrations in genome stability*".

We thank both referees and the editor of *Nature Communications* for giving us valuable advice and the opportunity to improve our manuscript. The reviewers' comments for final revisions are:

Reviewer #1 (Remarks to the Author)

Overall, the authors have addressed the comments made by the reviewers. There is one potential error I noticed in the supplementary figures with the circos plots for Chr1 and Chr2 potentially mixed up. There is a large loss/gain for Chr 1 in the main Figure, but this appears on Chr2 in the supplementary figure, which is potentially mislabelled.

We apologize for incorrect labeling of Chr1 and Chr2. We checked and corrected Supplementary Figure 4.

Reviewer #2 (Remarks to the Author)

The authors have appropriately addressed my concerns in the revised manuscript.